# COVID-19: The Course, Vaccination and Immune Response in People with Multiple Sclerosis: Systematic Review

**DOI:** 10.3390/ijms24119231

**Published:** 2023-05-25

**Authors:** Marcin Bazylewicz, Monika Gudowska-Sawczuk, Barbara Mroczko, Jan Kochanowicz, Alina Kułakowska

**Affiliations:** 1Department of Neurology, Medical University of Bialystok, M. Skłodowskiej-Curie 24A St., 15-276 Bialystok, Poland; 2Department of Biochemical Diagnostics, Medical University of Bialystok, Waszyngtona 15A St., 15-269 Bialystok, Poland; 3Department of Neurodegeneration Diagnostics, Medical University of Bialystok, Waszyngtona 15A St., 15-269 Bialystok, Poland

**Keywords:** COVID-19, SARS-CoV-2, antibodies, multiple sclerosis, vaccination, immune response

## Abstract

When the Coronavirus Disease 2019 (COVID-19) appeared, it was unknown what impact it would have on the condition of patients with autoimmunological disorders. Attention was focused on the course of infection in patients suffering from multiple sclerosis (MS), specially treated with disease-modifying therapies (DMTs) or glucocorticoids. The impact of Severe Acute Respiratory Syndrome Coronavirus 2 (SARS-CoV-2) infection on the occurrence of MS relapses or pseudo-relapses was important. This review focuses on the risk, symptoms, course, and mortality of COVID-19 as well as immune response to vaccinations against COVID-19 in patients with MS (PwMS). We searched the PubMed database according to specific criteria. PwMS have the risk of infection, hospitalization, symptoms, and mortality due to COVID-19, mostly similar to the general population. The presence of comorbidities, male sex, a higher degree of disability, and older age increase the frequency and severity of the COVID-19 course in PwMS. For example, it was reported that anti-CD20 therapy is probably associated with an increased risk of severe COVID-19 outcomes. After SARS-CoV-2 infection or vaccination, MS patients acquire humoral and cellular immunity, but the degree of immune response depends on applied DMTs. Additional studies are necessary to corroborate these findings. However, indisputably, some PwMS need special attention within the context of COVID-19.

## 1. Introduction

Multiple sclerosis (MS) is a chronic disease of the central nervous system (CNS), which affects nearly 2.8 million people worldwide [1]. The etiology of multiple sclerosis is complex and still unclear [2].

MS management includes three different methods. The first method relies on treating acute attacks (relapses), predominantly by administration of corticosteroids and plasmapheresis. The second method includes disease-modifying therapies (DMTs) with the use of immunomodulators and immunosuppressants. The third method is symptomatic treatment.

Considering the pathogenesis of MS and the use of DMTs, PwMS may be recognized as patients with immunodeficiency. Therefore, it has been suggested that the MS population is more exposed to the possibility of infection including higher susceptibility to viral infections than healthy people. Infection may also be a trigger factor that evolves into a specific phenomenon called pseudo-relapse [3].

In 2019, a previously unknown coronavirus in Wuhan, China was identified. The virus, called Severe Acute Respiratory Syndrome Coronavirus 2 (SARS-CoV-2), causes specific kinds of pneumonia with a rapid and traumatic course and is now known around the world as the Coronavirus Disease 2019 (COVID-19) [4]. As of 27 January 2023, 752,345,221 people were infected by SARS-CoV-2 of which 6,828,372 died. However, the precise number of MS patients who suffered from COVID-19 is still unknown [5]. COVID-19 may have different clinical presentations, from the asymptomatic course through mild symptoms such as cough, fatigue, headache, and diarrhea which occur within 2–14 days, up to serious acute respiratory distress syndrome (ARDS), or death. COVID-19 symptoms are dependent on, among other determinants, age, sex, and the patient’s immune status [6]. The infection of SARS-CoV-2 begins when the virus spikes receptor-binding domain (S-RBD) and reaches its specific receptors on the host cell, e.g., angiotensin-converting enzyme 2 (ACE2) or Neuropilin-1 (NRP-1) receptors [7,8]. Importantly, the specific receptors are presented not only in the respiratory system but also in other tissues and organs; therefore, coronavirus may also penetrate the nervous system. In COVID-19, the following diagnostics are used: reverse transcription polymerase chain reaction tests (PCR-RT) (the diagnostic gold standard) [9] and antigen and serological tests (antibody detection methods). The PCR-RT test detects SARS-CoV-2 RNA in patients’ fluids, such as nasopharyngeal swabs, saliva, sputum, and lower respiratory tract fluid [10]. Antigen tests focus on the detection of viral fragments which are involved in active viral infection [9]. Serological methods are also used to check if the patient has produced antibodies, which are synthesized when the patient comes into contact with the pathogen after infection or vaccination. Five classes of antibodies (IgM, IgA, IgG, IgD, and IgE) are known, but during the infection, the human immune system first produces IgM antibodies, which can begin inactivation of the virus after 5 days post-infection. IgG antibodies against the virus are detectable in human blood 3–5 days later [11]. Similar to IgM, IgA antibodies are involved early in the virus-neutralizing process, and it has been observed that this class of immunoglobulins is highly produced 8–14 days post-onset of symptoms [12]. Antibodies are present for a short time and some findings revealed that cellular immunity response, expressed in lymphocyte T activity, lasts longer than humoral response. It is known that T cells are activated by the Spike (S) protein domain of the SARS-CoV-2 structure [13]. It was shown that the immune response to Severe Acute Respiratory Syndrome Coronavirus-1 (SARS-CoV-1) could be present up to 17 years after contact with the pathogen. Performing similar observations of SARS-CoV-2 immune response could be an effective parameter of immune system activity against COVID-19 infection [14]. In fact, the immunological status of PwMS was greatly unknown, and it was unclear how this group would react to SARS-CoV-2 infection and which risk factors may lead to severe COVID-19 outcome/death in comparison to the general population. Several aspects were still unclear: how to perform DMT and relapse treatment, and how to obtain immunity in this group of patients and maintain it in the long term. Therefore, this review focuses on gathering and comparing scientific reports about the course of COVID-19, the immune response to SARS-CoV-2 infection, and vaccinations in PwMS treated by DMTs, and, furthermore, creating a complementary presentation of COVID-19 aspects in PwMS treatment.

## 2. Material and Methods

We performed a comprehensive literature search covering the period up to 30 January 2023. We used the MEDLINE/PubMed database with the following search strategy: keywords “COVID-19”, “Multiple Sclerosis”, “humoral immunity”, and “cellular immunity” in several variations, e.g., “COVID-19 AND multiple sclerosis” (1056 results). In the next step, we limited our search to studies published between 1 January 2019 and 30 January 2023, in English, and concerning humans. Then, the keywords “vaccination” and “treatment” were used, and we assessed for article eligibility. We excluded all non-significant papers (i.e., papers that did not contain information essential to the topic of our article or reports of single cases/small probe findings). Finally, 86 publications were included in this review (Figure 1. PRISMA Flow Diagram).

## 3. Results and Discussion

### 3.1. Risk, Course, and Mortality of COVID-19 in PwMS

Numerous studies report that the risk of SARS-CoV-2 infection in groups of PwMS is similar to the risk of infection in the general population. The Neuroimmunology Brazilian Study Group focused on COVID-19 and MS and compared a cohort of 11,560 MS patients to the general population in relation to the risk of infection. The study showed that the risk of SARS-CoV-2 infection was comparable in both groups and equal to 27.7/10,000 in the general population in comparison to 29.2/10,000 in MS patients [15]. Similar results were obtained by Solomon et al., on a 7000 PwMS probe. In the research, 1.1% of MS patients had positive SARS-CoV-2 tests. Comparatively, a positive test was noted in 2.5% of Toronto’s general population [16]. Naghavi et al., observed that 11.7% of 3050 PwMS had confirmed SARS-CoV-2 infection; while in the general Iranian population, infection was observed in 14.2% of citizens [17]. Evangelou et al., also noticed a similarity in the general population’s SARS-CoV-2 infection ratio. The PwMS were compared with their siblings and it was found that siblings without MS, in comparison with PwMS, are less likely to be infected by SARS-CoV-2 [18]. In comparison, Moreno Torres et al., observed the infection incidence ratio (0.78) in a group of 219 PwMS, which was lower than in the general population, but the hospitalization risk in PwMS was higher. The authors suggest that a higher hospitalization ratio could be caused by the fact that people with chronic diseases, especially those treated with immunosuppressive therapies, were more often admitted to hospitals than others. The investigation also showed a difference between the gender ratio of PwMS and people without MS. It has been revealed that females had a lower risk of infection than males in comparison to the general population. Male PwMS were also more likely to be hospitalized than females (1.5–3 times) [19]. It was observed that mortality among PwMS was the same or slightly lower than in the general population [19,20]. Similar results were shown by Sahraian et al., in a group of 4647 patients [21]. Prosperini et al., found that except for age and the presence of comorbidities, similar to the general population, two other risk factors are associated with a higher amount of death in the course of COVID-19: progressive MS course and management with DMT treatment (anti-CD20 drugs, interferon beta, and teriflunomide). What is more, they suggested that PwMS treated with anti-CD20 DMT and with progressive DMT should be especially prevented from SARS-CoV-2 infection. Contrarily, the authors did not find any correlation between mortality, male sex, and a higher degree of disability [22]. It is opposite to Sormani et al., findings which described a correlation between a higher degree of immobility, as expressed in the EDSS scale and COVID-19 mortality in PwMS. In addition, age was crucial to the chance of recovery. They observed that the median age of PwMS who died was 63 years, and the median age of the group who recovered was 45 years. According to Sormani et al., findings of elderly people with a high degree of disability need special attention in case of severe COVID-19 outcome risk [23]. Alroughani et al., reported a lower prevalence of COVID-19 in the MS group than in the general population and a lower risk of hospitalization. This was probably associated with younger age, absence of comorbidities in a great number of patients (90.3%), and low expanded disability status scale (EDSS) scores [24]. Additionally, Iaffaldano et al., compared two groups of patients: 779 PwMS infected by SARS-CoV-2 and 1558 PwMS without confirmed infection. The scientists observed that people who were infected were more frequently females, of younger age, treated with DMTs, and less disabled. Scientists suggested that it could be associated with a higher level of mobility and a higher number of contacts with other people, which has been translated into a higher chance of exposure to transmission of SARS-CoV-2 in this group [25]. Comparatively, patients with secondary progressive multiple sclerosis (SPMS) had a lower risk of infection than relapsing-remitting multiple sclerosis (RRMS) patients, which was probably caused by less frequent interpersonal contact or due to this group being treated less than the RRMS group [26].

According to NICE guidelines, when symptoms of COVID-19 infection are present 4–12 weeks after infection, it is called ongoing symptomatic COVID. If symptoms of infection are present after 12 weeks, and there is no other explanation, this state is called post-COVID [27]. Interestingly, Garjani et al., found that the MS population experienced prolonged COVID-19 symptoms (30% in 4 weeks and 12% in 12 weeks) than the general population (13% and 2%, respectively) [28].

During COVID-19, people in the general population suffered mainly from fever, pneumonia, and dry cough. Several authors also mentioned other symptoms such as fatigue, headache, anosmia/hyposmia, muscle pain, shortness of breath, and myalgias/arthralgias in various types of combinations. It was shown that PwMS symptoms are similar to those of the general population. Thus, it seems that the manifestation of SARS-CoV-2 infection does not depend on whether the person suffers from MS, but rather on the region where they live and personal conditions [19,29,30]. The very important question to answer is whether the course of SARS-CoV-2 infection is also similar to that in the general population and which factors can influence the course of the disease. Many authors describe the course of COVID-19 as asymptomatic, mild, moderate, or severe. However, Alroughani et al., described a mild course of COVID-19 as not requiring hospitalization, moderate when hospitalization is needed, and severe when intensive care is needed [24]. Most PwMS (88.8% up to 96% of cases) suffered from mild COVID-19 [15,16,31,32]. It may be associated with young age and the absence of comorbidities in most PwMS [24,31]. In the Safavi et al., study, the research method was based on the 2000 surveys sent to MS patients with a COVID-19 diagnosis. Scientists observed that in MS the course of SARS-CoV-2 infection was mild or moderate. Moreover, Safavi et al., suggested that the severe course of COVID-19 may be less frequently noted simply due to the inability to complete the questionnaire by those patients with severe conditions [31]. Alonso et al., suggest that older age, higher disability, and longer MS duration are associated with COVID-19 severity, but there is no compatibility with gender, obesity, or DMT treatment [33]. The correlation between disability and severe COVID-19 risk was found by scientists who observed that impairment of expiratory muscles and low cough efficiency, which is common in more disabled PwMS, may lead to an increased risk of severe COVID-19 outcome [34].

### 3.2. Multiple Sclerosis Treatment and COVID-19

The utilization of DMT may be a crucial factor affecting the risk of severe SARS-CoV-2 infection course in PwMS. Most DMTs seem to not influence risk, outcome of infection, and response to vaccinations. Interestingly, research performed by Freedman et al., revealed that IFN β-1a-treated MS patients had a lower risk of severe course of COVID-19 and mortality than the general population and population of MS patients without INF IFN β-1a treatment [35]. Landi et al., found that natalizumab-treated patients had no association with worse or prolonged outcomes, but their research is limited by a small sample size of only 18 MS SARS-CoV-2 infected patients [36]. Another investigation confirmed that fingolimod and siponimod were not involved in a higher risk of severe COVID-19 outcomes in comparison with the general population [37]. There are some concerns about anti-CD20 monoclonal antibodies and sphingosine-1-phosphate (S1P) therapies. Reder et al., have shown an increased risk of infection in PwMS treated with anti-CD20 monoclonal antibodies, and a decreased risk in groups treated with interferons or glatiramer acetate [38]. These findings were supplemented by Simpson-Yap et al., who found that rituximab and ocrelizumab therapy is associated with a higher risk of severe COVID-19 outcome in comparison to treatment with dimethyl fumarate [39].

Additionally, Sormani et al., observed that people treated with anti-CD20 drugs (ocrelizumab and rituximab), people of older age, males with present comorbidities, people with a higher disability level, or people suffering from MS for a long period of time, had a higher probability of severe COVID-19 outcome [23]. A comparison of SARS-CoV-2 infection risk and hospitality rate depending on the treatment was extracted in Table 1.

Spelman et al., also found that rituximab treatment relates to a higher risk of severe COVID-19 outcome, and they suggested a significant role of B cells in the immune response against SARS-CoV-2 infection [40]. However, McKay et al., were not able to find any statistically significant association between the timing of infusion, cumulative dose of rituximab, and COVID-19 hospitalization risk [41]. Moreno-Torres et al., did not find an association between DMT and glucocorticoid treatment and COVID-19 outcome [19]. Similarly, Czarnowska et al., did not discover any significant difference in the severity of SARS-CoV-2 infection in relation to age, MS duration, disability level, comorbidities, or DMT treatment [42]. Similar conclusions were reported in the articles of Alshamrani et al., and Sormani et al., but interestingly, they reported an increased risk of severe COVID-19 course up to 4 weeks after glucocorticoid treatment [23,43]. What is more, Naser Moghadasi et al., have found that there is no significant association between the possibility of SARS-CoV-2 infection and intravenous glucocorticoid therapy. However, they revealed that prolonged pulse glucocorticoid therapy may increase the risk of COVID-19 infection [44].

### 3.3. Vaccination and Immune Response

In 2020, vaccines against COVID-19 were developed. There are three types of COVID-19 vaccines worldwide. The mechanism of the first vaccine is based on mRNA implementation that codes Spike protein (BNT162b2, mRNA-1273) [45,46]. The second type of vaccine relies on whole Spike SARS-CoV-2 protein introduction (NVX-CoV2373) [47]. The third type of vaccine uses a vector (adenovirus) that carries genetic material to cells (ChAdOx1 nCoV-19, Ad26.COV2.S) [48,49]. A special group of people with autoimmune diseases, such as MS, were vaccinated [46]. It was suggested that people with MS should undergo vaccination because the complications that could occur during SARS-CoV-2 infection seemed to be more serious than the vaccination side effects [50]. Data from the available literature suggest that the percentage of vaccinated PwMS is 70.6%–90.0%. Moreover, most PwMS received the mRNA vaccine [51,52,53,54]. Differences in the percentage of PwMS may result from a different time when the research was conducted, regional health system organization, or patients’ personal willingness to be vaccinated.

Some side effects may be associated with vaccinations against SARS-CoV-2. Symptoms may occur after the first or second dose of vaccine, shortly after injection, and usually remain for a short period of time [55,56]. In PwMS, similar to the general population, the most common adverse events were mild; the most common were injected arm pain, flu-like symptoms, fatigue, and headache [55,56,57]. Interestingly, the type of adverse events after the first and second doses were similar, and their frequency after the second dose was slightly lower than the first dose [55]. Briggs et al., also found that PwMS, especially younger females after SARS-CoV-2 infection, had a higher risk of adverse events after the first dose of vaccine (ChAdOx1 nCoV-19 or BNT162b2) [57]. In opposition, Lotan et al., observed that MS patients had a lower rate of adverse events than the general population [56]. Another study, by Dreyer-Alster, found that 54.5% of MS patients who were given a third dose of vaccine against SARS-CoV-2 had adverse events similar to that after the first and second doses. They also observed that 3.3% of patients who had MS relapse needed intravenous glucocorticoid treatment [58].

Zabalza et al., found that 3 months after SARS-CoV-2 infection, 45.6% of PwMS had positive results of anti-SARS-CoV-2 IgG antibody presence and that the concentration increased with the severity of the COVID-19 course. They also discovered significance with anti-CD-20 therapy and lower positive antigen test percentage, compared with other DMT treatments or even without it (17.6% vs. 48.8% and vs. 68.4%). They noticed that serological response in PwMS without DMT treatment was higher than in the group treated with DMTs. They suggested that immunization through seronegative subjects could probably be expressed in another way, such as T cell response [59]. Van Kempen et al., drew similar conclusions [60]. Bsteh et al., also assessed the level of antibodies in PwMS about 5 months after infection. They observed that anti-SARS-CoV-2 antibody titers were slightly lower in the group treated by immunosuppressive DMTs compared to patients treated with immunomodulatory DMTs or without DMT. However, similar to the general population, there was no relevant relationship between immunological response and age, sex, and severity of COVID-19 [61].

Iannetta et al., performed an investigation on PwMS treated with ocrelizumab (an anti-CD-20 drug of at least one year of treatment) who suffered from COVID-19. They discovered that those people had low levels of B cells and normal T cells subset in blood [62]. It has been suggested that humoral response to SARS-CoV-2 infection in the MS group may be blunted after anti-CD-20 treatment [63]. Therefore, measuring T cell response by SARS-CoV-2 specific IGRA test may be useful in the MS population [62].

In most research papers, it is noted that this kind of immunization is suggested for all patients with diagnosed multiple sclerosis [64]. It was shown that there is no correlation between vaccination and activity of MS expressed through EDSS progression and count of acute relapses (approximately 2%), which is important because infections increase the risk of occurrence of relapse or pseudo-relapse [64,65,66]. It was shown that most of the DMTs do not decrease immunological response to vaccinations against COVID-19. A study performed by Capone et al., revealed similar humoral response in MS patients treated with interferon, glatiramer acetate, dimethyl fumarate, teriflunomide, and natalizumab in comparison to the healthy general population [67]. Similar results were presented in a 6-month observation study performed by Altieri et al., on people with multiple sclerosis who were treated with natalizumab [68]. In addition, a reduced humoral response was not observed in groups of PwMS treated with alemtuzumab and cladribine. Unfortunately, the research groups were limited, and long-term conclusions require additional studies [67].

Another study revealed that type 1 interferons, glatiramer acetate, and teriflunomide seem to not blunt vaccines against COVID-19 effects [65]. Additionally, Pitzalis et al., performed research on a Sardinian group and found that natalizumab, teriflunomide, and azathioprine rituximab indicated a lower humoral response; dimethyl fumarate, interferon, alemtuzumab, and glatiramer acetate had no significant difference between the general population [69].

However, some findings revealed that there are DMTs that may decrease post-vaccination response against COVID-19. Interesting research was performed by Tortorella et al., 2–4 weeks after the second vaccination dose, the group of PwMS with DMT treatment was compared with healthcare workers. It was revealed that PwMS treated with ocrelizumab, fingolimod, and cladribine had a lower humoral immunity response than healthcare workers and PwMS treated with interferon-beta. T cells’ reactivity in response to vaccination was measured by the level of released interferon-gamma. Most of the participants showed T cell response (62%); although it was lower in the group of PwMS than in the group of healthcare workers. The lowest level of cellular immunity was observed in a group of PwMS treated with fingolimod (14.3%) [70].

Tallantyre et al., and Krbot Skorić et al., found that anti-CD20 therapy, and fingolimod and siponimod (sphingosine-1-phospohate receptor modulator-S1P) therapies, caused lower levels of seroconversion in opposition to other DMTs. Moreover, fingolimod-treated people had lower T cell counts. Importantly, 40% of seronegative subjects had positive T cell response. There was also a suggestion that MS patients for whom treatment with anti-CD 20 and fingolimod was planned, should first undergo vaccination, and then treatment should be performed [71,72]. Kister et al., obtained similar results regarding therapy with siponimod, measured by INF-gamma level, but they also reported increased T cell reactivity in PwMS with natalizumab treatment. Furthermore, no race and ethnicity correlation was revealed [73]. Another study revealed that a 5-month interval between ocrelizumab administration and vaccination against SARS-CoV-2 might result in the presence of higher levels of IgG antibodies and, in effect, a larger amount of seropositive PwMS. This observation may be very important in the context of better protection against COVID-19 severe outcomes in PwMS treated with ocrelizumab [74]. Moreover, according to König et al., observations, the anti-CD20 group should probably be administered an additional dose of vaccination. They observed that people treated with anti-CD20 drugs had slightly increased anti–SARS-CoV-2-spike-RBD IgG antibodies after revaccination (after three doses) and it was significantly higher than in the group of patients treated with fingolimod [75,76]. In another study of treatment with ocrelizumab and fingolimod in MS patients, the humoral response was measured slightly earlier (mean 33.1 days after application of two doses) than in Zabalza et al., or Bsteh et al., investigations [59,61]. Treated with fingolimod, the MS group reached 62.5% positive serological results whereas ocrelizumab was 37.5%. The authors suggested that, depending on therapy, antibody levels could be alterable [77]. Another study revealed that anti-SARS-CoV-2 IgG antibodies were detected in approximately 75% of PwMS after the second and third doses of the BNT162b2 mRNA vaccine, but Milo et al., also confirmed that the level of seroconversion in PwMS treated by ocrelizumab and fingolimod was lower (accordingly, ocrelizumab: 38% and 44%; fingolimod: 54% and 46%). A similar phenomenon was observed in PwMS treated with ocrelizumab and fingolimod, but after having undergone COVID-19 [74]. It was also observed that discontinuing MS fingolimod treatment may help to develop humoral immunity against SARS-CoV-2 infection. Interestingly, there were some cases where a humoral response was absent after vaccination against SARS-CoV-2. Achiron et al., studied a group of PwMS treated with fingolimod with no humoral response after obtaining two doses of BNT162b2-mRNA; they divided the group—one part of the group continued therapy, and the other part of the group had the fingolimod changed. Then, both groups were vaccinated with a third dose of the COVID-19 vaccine. Next, humoral and cellular immunity markers were measured 1 and 3 months after the third vaccination. It was revealed that the group whose DMT treatment was modified had a higher percentage of anti-SARS-CoV-2 IgG antibody presence and higher IgG concentrations (202.3 BAU/mL vs. 26.4 BAU/mL). They also found a correlation between IgG humoral response and lymphocyte count. Unfortunately, no cellular memory markers were present after the third COVID-19 vaccination was observed [78]. Tallantyre et al., revealed that PwMS treated by DMT were seronegative after two doses of vaccination against SARS-CoV-2. However, scientists suggest that after vaccination with a booster dose, one-third of patients could be seroconverted and approximately nine in ten may have a detectable immune response against SARS-CoV-2. They also revealed that PwMS with prior vaccination of CHAdOx1 nCoV-19 had a greater chance of seroconverting than with the BNT162b2-mRNA vaccine [79]. This may have an important impact on the choice of vaccine type to use in the case of PwMS treated with anti-CD20 DMT.

Interesting results were obtained by Habek et al., who compared groups of PwMS treated with ocrelizumab, treatment-naïve, or treated with another first-line DMT with healthy controls, after undergoing COVID-19 and/or vaccination against SARS-CoV-2. Humoral and cellular immunity was measured by the level of SARS-CoV-2 IgG antibodies and values of CD4 and CD8 cells expressing INFγ, TNFα, and IL2. It was observed that PwMS treated with ocrelizumab had a lower humoral immune response after undergoing COVID-19 as well as vaccination. However, the levels of cellular immunity in both groups were similar. Unfortunately, this study was limited by a small number of participants [80]. Another result was obtained by Milo et al., who found that cellular immunity expressed in INFγ T cell activity in PwMS treated with ocrelizumab after the second and third doses of the COVID-19 vaccine was higher than in PwMS treated with other DMTs, notably in comparison with fingolimod treated patients (respectively, 100%, 85% ocrelizumab; 25%, 0% fingolimod; and 89%, 63% all PwMS). In this study, blood was collected 6 months after obtaining the second dose, and 1–14 weeks after the third dose, of the BNT162b2 mRNA vaccine [74]. What is more, Zabalza et al., found that in PwMS treated with anti-CD20, the cellular immunity response was present up to 13 months after COVID-19, even if humoral immunity was absent. It was also found that humoral immunity against SARS-CoV-2 was enhanced in males whereas cellular response was decreased in PwMS cases who had a progressive course of illness [81]. However, Satyanarayan et al., presented that PwMS treated with ocrelizumab may have a greater chance of obtaining humoral response after vaccination against SARS-CoV-2 if they have previously undergone COVID-19 infection or if they have a shorter therapy duration [82]. Regrettably, the data about cellular immunity against SARS-CoV-2 infection or vaccination, with particular emphasis on long-term observations, are still limited and should be updated.

Unfortunately, the minimum level of antibodies required to protect from COVID-19 severe outcomes is not known [83]. Sormani et al., discovered that a 10-fold increase in antibody levels is associated with a 43% decreased risk of infection. They also observed that a level ≤ 659 BAU/mL is associated with a higher risk of infection up to 6 months. When the Omicron COVID-19 variant appeared, they noticed a six times higher risk of infection. Vaccinations decreased the risk of hospitalization from 12.8% in the MS group before vaccination to 3.1% after immunization [84]. Importantly, it was found that in the case of PwMS, mRNA vaccines are more effective at developing a humoral immune response against COVID-19 than non-mRNA vaccines [85].

## 4. Conclusion

The COVID-19 pandemic had a large impact on many aspects of human life, especially in people suffering from chronic autoimmunological diseases such as multiple sclerosis. This review focuses on the course of COVID-19, the immunological response to SARS-CoV-2 infection, and vaccinations against COVID-19 in PwMS. PwMS have a risk of infection, hospitalization, and mortality similar to the general population. However, it was shown in some studies that patients treated with anti-CD20 antibodies and fingolimod are likely to have worse COVID-19 outcomes and vaccination response than PwMS treated with other DMTs. Vaccination against COVID-19 in PwMS treated with anti-CD20 antibodies and glucocorticoid administration in case of relapse should receive special attention. Vaccinations against COVID-19 are well tolerated and strongly recommended for all populations suffering from multiple sclerosis, especially those who have COVID-19 severe outcome risk factors such as male gender, older age, presence of comorbidities, higher EDSS degree, or longer MS duration. The information included in this review may be useful for physicians who treat PwMS with DMT or manage relapse and need to obtain information about the COVID-19 course, vaccinations against COVID-19 administration, or the development of humoral and cellular immunity in this group of patients. It will also help spread information about the safety of vaccinations and their proper administration. This review has some limitations: one database research (MEDLINE/PubMed database); limited information, e.g., limited epidemiology data on the worldwide occurrence of COVID-19 in PwMS; the sparse amount of information on vaccination against COVID-19, divided by type; and limited observation of long term humoral and cellular immunity. We hope that further investigation may deliver more data. Long-term follow-ups are necessary to draw proper conclusions about immune system activity in vaccinated and non-vaccinated PwMS; indisputably, the data about the immunological response to the vaccination against SARS-CoV-2 in PwMS are still limited.

## Figures and Tables

**Figure 1 ijms-24-09231-f001:**
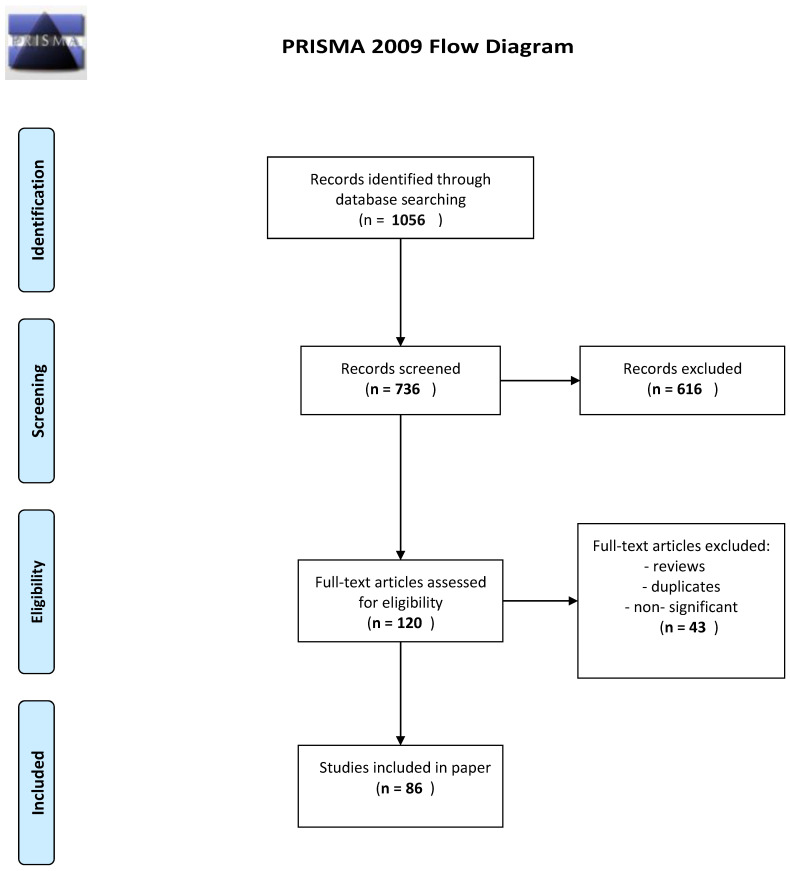
PRISMA Flow Diagram.

**Table 1 ijms-24-09231-t001:** SARS-CoV-2 infection risk and COVID-19 hospitalization rate PwMS with/without DMT treatment *.

DMT	SARS-CoV-2 Infection Risk	COVID-19 Hospitalization Rate	References
Anti-CD20	higher (vs. PwMS with DMT treatment)	higher (in PwMS with DMT treatment)/ lower (vs. PwMS without DMT treatment)	[35,36]
Anti CD52	higher (vs. PwMS with DMT treatment)	lower (in PwMS with DMT treatment)/ lower (vs. PwMS without DMT)	[35,36]
Anti-VLA-4	higher (vs. PwMS with DMT treatment)	lower (in PwMS with DMT treatment)/ lower (vs. PwMS without DMT)	[35,36]
Fumarate dimethyl	slightly higher (vs PwMS with DMT treatment)	higher/ lower(in PwMS with DMT treatment)/ lower (vs PwMS without DMT treatment)	[35,36]
IFNβ	lower (vs. PwMS with DMT treatment)	higher (in PwMS with DMT treatment)/ lower (vs. PwMS without DMT treatment)	[35,36]
Glatiramer acetate	lower (vs. PwMS with DMT treatment)	higher/ lower (in PwMS with DMT treatment)/lower (vs. PwMS without DMT)	[35,36]

* PwMS = People with Multiple Sclerosis; DMT = Disease Modifying Therapy.

## Data Availability

No new data were created or analyzed in this study. Data sharing is not applicable to this article.

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
