# Peer review of "COVID-19: The Course, Vaccination and Immune Response in People with Multiple Sclerosis: Systematic Review"

_ijms, 2023, doi:10.3390/ijms24119231_

Round 1
Reviewer 1 Report
The authors wrote a systematic review focusing covid 19 in people li I got with multiple sclerosis. The paper is interesting and well organized. But I have some major concerns:
1 the introduction should focus more on covid in me patients, in particular on what is already know (risk factors for severe covid 19, effect of me x de on covid 19 severity, vaccination issues, etc) , on what is not know and the major knowledge gaps to cover . At last the authors should better explain the objectives of the paper.
materials and methods are weak: the reader should be able to replicate the literature search and selection after reading this section. Please better explain Selection criteria in each single phase.
the authors should discuss some other issues : covid 19 mortality in ms: there are a couple of paper of interest (Sormani putting data in right context and Prosperini Determinants of COVID-19-related lethality in multiple sclerosis: a meta-regression of observational studies. Further more they should explain disability as a risk factor for severe the pneumonia (immovilli lung .. correlates …)
In the discussion they should analyze how to globally approach covid 19 in people with ms (immovilli … a risk to benefit approach).
English is fine
Reviewer 2 Report
I would like to thank the authors and the Editorial Board for the opportunity to review the article submitted to International Journal of Molecular Sciences. I believe that the presented review bring new and important information for both clinical practitioners and scientific researchers. In my opinion, some minor changes could be implemented in order to improve the quality of the reviewed manuscript. Below I present my comments on the individual sections of the authors’ article.
Title: The current title does not reflect the aim of the study. Authors decided to gather information about the course of COVID-19, immune response and vaccinations in MS patients. In my opinion it should be specified in the title, because in the current form, it suggest that the article could also reflect on the psychological functioning in MS patients’, which is not present there.
References: In my opinion, some references are off-topic . The aim of this study is to show the results of a worldwide systematic review. In lines 29-30 the authors write “MS (…) affects nearly 2.8 million people worldwide [1], and 45,000 in Poland [2]”. What’s the point of including statistics of Polish individuals, since the presented review takes into account results from the whole world? Some of the authors of the reviewed manuscript did also co-author the reference number 2, and this reference does not bring anything in line with the aim of the reviewed manuscript. Therefore, it could be interpreted as an example of self-citation farming. There is nothing wrong in referring to one’s own work when it is in-line with the aim of the written manuscript – unfortunately, it is not always the case in here. There are multiple examples of the self-referencing in the presented manuscript. I highly recommend that both the authors of the reviewed manuscript and MDPI’s assigned Editor review the literature presented in this manuscript and remove all off-topic references.
Introduction: The presented literature shows important information on MS and COVID-19 by itself, but does not refer to epidemiological data of the COVID-19 in MS patients. I highly recommend that the authors refer to the data showing, how many individuals suffered from COVID-19 worldwide and how many of them were MS patients.
Methods: Please elaborate, why only PubMed database was used and why the authors decided to exclude other databases such as Scopus or Web of Science, which sometimes index additional records which are not available via PubMed.
Methods: Please explained, why only two keyword combinations were used. In other words, why other synonyms were not used, such as SARS-CoV-2 or sclerosis multiplex. This probably highly impacted the amount of the gathered records.
Methods: Please describe, what criteria were used to exclude “small probe findings” (line 85). Initially the authors included 1056 publications, 736 articles were screened, and 616 were excluded – those numbers are very high, therefore the used methodology should be described in-depth.
Results: I highly recommend that the authors explain why the systematic review was performed instead of a higher-tier meta-analysis approach. If meta-analysis was not possible, please state those reasons.
Discussion: Please expand the discussion section. It lacks important information on: (1) this review’s limitations; (2) practical implications of the presented results
